# Adaptation and Validation of the Nova-UPF Screener for the Assessment of Ultra-Processed Food Intake in Portuguese Adults

**DOI:** 10.3390/nu18010090

**Published:** 2025-12-27

**Authors:** Sandra Abreu, Caroline dos Santos Costa, Margarida Liz Martins

**Affiliations:** 1School of Life and Environment Sciences, University of Trás-os-Montes and Alto Douro (UTAD), 5001-801 Vila Real, Portugal; 2RISE-Health, Department of Genetics and Biotechnology, School of Life and Environmental Sciences, University of Trás-os-Montes and Alto Douro, 5000-801 Vila Real, Portugal; 3Research Centre in Physical Activity, Health and Leisure (CIAFEL), Faculty of Sport, University of Porto, 4050-313 Porto, Portugal; 4Laboratory for Integrative and Translational Research in Population Health, 4050-600 Porto, Portugal; 5Centre for Epidemiological Studies in Nutrition and Health, School of Public Health, University of São Paulo, São Paulo 01246-904, Brazil; carolinercosta@gmail.com; 6Coimbra Health School, Polytechnic University of Coimbra, 3045-093 Coimbra, Portugal; margarida.liz@estesc.ipc.pt; 7H&TRC—Health & Technology Research Center, Coimbra Health School, Polytechnic University of Coimbra, 3045-093 Coimbra, Portugal; 8Sports and Physical Activity Research Center, University of Coimbra, 3040-248 Coimbra, Portugal; 9Research Centre for Anthropology and Health, University of Coimbra, 3000-456 Coimbra, Portugal

**Keywords:** food classification systems, NOVA classification, ultra-processed foods, validation study

## Abstract

**Background/Objectives**: With the increasing global concern about diet-related diseases associated with the consumption of ultra-processed foods (UPFs), there is an urgent need for practical and standardized tools to evaluate and monitor UPF intake. This study aimed to adapt and validate the Nova-UPF screener, a brief, food-based questionnaire originally developed in Brazil, for use among Portuguese adults. **Methods**: The adaptation process relied on data from the Portuguese National Food, Nutrition and Physical Activity Survey (IAN-AF 2015–2016) and was refined using the DELPHI methodology. A validation study was carried out with a convenience sample of 304 adults through an electronic questionnaire. Dietary intake was evaluated using a 24 h dietary recall. Criterion validity was assessed by examining the relationship between the Nova-UPF score and the percentage of total energy intake (TEI) from UPFs, while construct validity was evaluated based on predefined hypotheses. Agreement between quintiles of Nova-UPF score and quintiles of UPF contribution to TEI was tested using the prevalence and bias-adjusted kappa (PABAK) index. **Results**: The final Portuguese version of the Nova-UPF screener includes 25 subgroups. The Nova-UPF score was positively associated with the percentage of UPF contribution to TEI (B = 6.224, *p* < 0.001). Participants in the highest quintiles of the Nova-UPF score had higher TEI but lower consumption of monounsaturated and polyunsaturated fats, potassium, and dietary fibre. There was a near-perfect agreement between quintile classifications of UPF and Nova-UPF score (PABAK = 0.86). **Conclusions**: The Portuguese Nova-UPF screener is a valid, simple, and quick tool for evaluating UPF consumption and dietary quality in adults.

## 1. Introduction

Changes in the dietary patterns of populations in recent decades has determined an increase in consumption of “convenience” foods and beverages, ready-to-eat products and packaged food that replaced traditional dietary patterns based on minimally processed foods and freshly and home-prepared meals [1,2]. The consumption of ultra-processed foods (UPFs) has grown exponentially in recent decades, becoming a dominant feature of global food systems. UPFs are characterized by their industrial formulations, containing minimal or no whole foods, and are often high in added sugars, unhealthy fats, salt, and artificial additives [3]. These foods are designed for convenience, long shelf life, and hyper-palatability, making them highly appealing but nutritionally poor [3].

Considering this, the impact of UPFs consumption on the quality of diets and health outcomes has gain relevance. Scientific evidence has been shown the association between UPFs consumption and non-communicable diseases, namely overweight and obesity, also suggesting a probable relationship between UPF consumption and other health outcomes, such as cardiovascular disease, type 2 diabetes, gastrointestinal disorders, and cancer [4,5,6]. A recent landmark analysis concluded that the displacement of long-established dietary patterns by UPFs is a key driver of the escalating global burden of multiple diet-related chronic diseases [7].

Furthermore, UPFs are associated with poor dietary quality, contributing to the displacement of minimally processed and nutrient-dense foods [8,9]. Lauria et al. [10] showed that most of the daily energy intake of European consumers derives from UPFs for different age and socio-economic groups. This study developed in eight European countries has also presented that an increase in the consumption of UPFs is associated with unhealthy dietary patterns characterized by high consumption of sugar, total fat and saturated fat, and low consumption of protein and fibre [10].

In Portugal, the UPFs availability between 1990 and 2005 increased from 3.9% to 13.8% [1]. Data from the National Food, Nutrition and Physical Activity Survey of the Portuguese Population (IAN-AF 2015–2016) showed that a higher consumption of UPFs in adults was associated with higher total energy, dietary energy density, higher content of free sugars, total fats and saturated fats, and lower dietary density of fibre, sodium and potassium and protein content [9]. UPFs contributed to, approximately, 24% of the energy consumed by adults. Younger adults (18–44 years) presented the highest consumption of UPFs (340 g/day), which have been associated with the worst nutrient profile dietary pattern, poor in most vitamins and minerals [11,12].

Traditionally, the assessment of UPF consumption is conducted through food diaries, 24 h recall questionnaires, or food frequency questionnaires. These methods are costly, time-consuming for both respondents and interviewers, and complex to analyze [13,14]. Given the complexity of dietary assessment, it is essential to find alternative instruments to evaluate and monitor UPFs consumption.

In order to achieve these goals, Costa et al. [15] developed and validated the Nova-UPF screener, a simplified instrument to assess UPF consumption among Brazilian adults. This tool includes questions about the previous day’s dietary intake of a variety of subgroups of UPFs, which provides the Nova-UPF score for the consumption of UPFs. It is a simple and quick tool to administer, requiring minimal effort, user-friendly, and capable of facilitating the monitoring and evaluation of UPF consumption, as well as its potential impact on the development of diet-related non-communicable diseases [15]. The implementation of such practical and standardized assessment tools is in line with World Health Organization (WHO) recommendations emphasizing the need for systematic surveillance of UPF intake, given its growing association with obesity and other diet-related chronic diseases across the European region [16]. Considering this and the importance of easily assessing UPFs consumption in the Portuguese population, it is an important issue to adapt and validate the Nova-UPF screener.

The Nova-UPF screener tool has already been adapted and validated for the adult population of Senegal [17], Ecuador [18], and India [19], as well as for young women aged 14–20 years old in the city of Medellín (Colombia) [20], proving to be a valid and accurate tool for measuring and monitoring the consumption of UPFs at the population level in these countries.

This study aims to adapt and validate the short food-based screener (Nova-UPF screener) developed by Costa et al. [15] for the Portuguese adult population. By ensuring its cultural and linguistic adaptation, Nova-UPF screener will enable an accurate assessment of UPF consumption. This, in turn, will provide valuable insights for public health initiatives and policymakers aimed at reducing UPFs consumption and promoting healthier dietary patterns in Portugal.

## 2. Materials and Methods

### 2.1. Ethics

This study was conducted in accordance with the Declaration of Helsinki and approved by the Ethics Committee of the University of Trás-os-Montes and Alto Douro (protocol code Doc56-CE-UTAD-2022). Written informed consent was obtained from all participants prior to their inclusion in the study.

### 2.2. Adaptation of the Nova-UPF Screener

The Nova-UPF screener was originally developed and validated for Brazilian adults as a brief, food-based screening tool designed to assess the consumption of UPFs, rather than to provide a quantitative estimate of UPF intake in grams or kilocalories [15]. The primary purpose of the tool is to enable rapid assessment and monitoring of UPF consumption, as using more complex dietary assessment methods can be expensive and time consuming. The screener includes a list of 23 UPF subgroups divided into three main categories: beverages (six subgroups), meal replacements or accompaniments (ten subgroups), and products typically consumed as snacks (seven subgroups). Participants are asked to indicate whether they consumed any item from each subgroup on the previous day. Each subgroup consumed receives a score of 1, while non-consumption receives a score of 0. The total Nova-UPF score is obtained by summing the number of UPF subgroups consumed, resulting in a score that reflects the frequency of UPF exposure, with higher scores indicating greater UPF consumption.

The adaptation of the subgroups included in the Portuguese version was based on national data from the IAN-AF 2015–2016 [9]. A panel of 11 experts in public health nutrition, epidemiology, and the food industry was invited to assess the content validity of the initial draft of the adapted tool for the Portuguese context, ensuring gender and educational equalities. Following the Delphi methodology [21], an online survey was conducted to gather expert feedback on each subgroup. In each round, experts evaluated the relevance, clarity, and cultural adequacy of the proposed UPF subgroups. After the first round, an anonymized summary of the group responses and justifications was shared with all panel members. They were then invited to revise their initial judgments considering the group feedback. Consensus was defined a priori as agreement by at least 75% of the experts on each item. All items reached the predefined level of consensus by the end of the second round, at which point the process was concluded according to the predefined stopping criterion. Final decisions regarding item inclusion, modification, or exclusion were based on the median ratings of the final Delphi round. The Nova-UPF screener was subsequently revised and adjusted based on expert input to produce the final version of the instrument.

### 2.3. Face-Validity

To assess the face validity of the Portuguese version of the Nova-UPF screener, a pre-test was conducted with a convenience sample of 10 adults aged 18–65 years. Participants were asked to complete the questionnaire and provide feedback regarding the clarity, comprehensibility, ease of interpretation, and relevance of the items. Specifically, participants were asked to comment on (i) whether the food items or groups were clearly written; (ii) whether the meaning of each item was easily understood; (iii) whether the brand names included were familiar and recognizable; (iv) whether the listed items were relevant to their usual diet; (v) whether any commonly consumed foods or beverages similar to those listed were missing; and (vi) whether any foods on the list were never or very unlikely to be consumed. Based on participant feedback, minor wording (e.g., replacing “surimi” with the more culturally familiar term “delícias do mar”), and formatting (e.g., bold and capital letters for the recall period), adjustments were made to improve clarity and ensure cultural appropriateness for the Portuguese population.

### 2.4. Data Collection and Procedure for Validation Study

#### 2.4.1. Study Population

This study included a convenience sample of individuals aged between 18 and 65 years old. Health centres, private institutions for social solidarity, local authorities, and academic institutions were contacted to participate and disseminate information about the study. Individuals were then invited either via institutional email or verbal communication (personal announcements and word-of-mouth). Subjects who were aged 18–65 years at the time of recruitment were included in the study. Individuals were ineligible if they were unable to speak, understand, and write Portuguese, or if they had dementia or another mental condition that made them incapable of filling in questionnaires properly.

The sample size was determined to detect a correlation coefficient of ≥0.30 at a significance level of 0.05, two-sided tails and considering a power of 80%. The required sample size was 84, assuming a drop-out rate of 20%, meaning we would need 101 individuals.

#### 2.4.2. Data Collection

Data collection was conducted between November 2023 and March 2024 by four trained researchers who underwent a one-day training session to ensure consistency in data collection. Each participant was informed about the study objectives and, after providing consent, completed a socio-demographic and lifestyle questionnaire using an electronic data form (Google Forms^®^). Participants then independently filled out the Nova-UPF screener, which included a list of UPF subgroups, indicating all items they had consumed the previous day. After completing the Nova-UPF screener, a 24 h dietary recall interview was administered, during which participants reported all foods and beverages consumed, along with the corresponding quantities.

#### 2.4.3. Sociodemographic and Lifestyle Information

Sociodemographic factors such as age, educational level, sex, marital status (single/divorced and married), and urbanization degree, as well as lifestyle measures like dietary pattern, smoking, and physical activity, were assessed through a self-administered questionnaire. Educational level was categorized based on divisions within the Portuguese educational system: basic (≤9 school years), secondary (10–12 school years), and college/university (>12 school years).

The International Physical Activity Questionnaire Short-Form (IPAQ-SF) was used to assess physical activity. The IPAQ-SF has been previously validated and adapted for the Portuguese population. It comprises 7 items with open-ended questions about individuals’ physical activity in the last 7 days [22].

Information on tobacco use was self-reported by participants and categorized into four groups: non-smokers, former smokers (those who had quit smoking for at least six months), occasional smokers (those who smoked, on average, less than one cigarette per day), and current smokers (those who smoked at least one cigarette per day) [23]. Since occasional smokers accounted for only a small proportion of the sample (4.8%), they were recoded and combined with current smokers for analysis.

Participants were asked to self-report their body weight and height. Body mass index (BMI) was calculated by dividing weight by height squared (kg/m^2^). According to the WHO classification [24], BMI values were categorized as follows: underweight (<18.5 kg/m^2^), normal weight (18.5–24.9 kg/m^2^), overweight (25.0–29.9 kg/m^2^), and obesity (≥30.0 kg/m^2^). For analytical purposes, due to the small number of participants classified as underweight (3.5%) they were combined with normal weight individuals as non-overweight. Similarly, obese participants (5.1%) were combined with overweight individuals.

#### 2.4.4. 24 h Dietary Recall

The 24 h dietary recall was conducted using the five-stage multiple-pass method to enhance the completeness and accuracy of dietary reporting [25]. These steps included (1) asking participants to quickly and uninterruptedly report all foods and beverages consumed during the previous day; (2) prompting them with a forgotten foods list to aid in recalling commonly omitted items; (3) inquiring about the time and occasion of consumption for each food item; (4) obtaining a detailed description and estimating portion sizes; and (5) conducting a final probe review to verify and correct any inconsistencies. This standardized approach minimizes underreporting and improves data quality in dietary assessment. Portion sizes were estimated using the Photographic Manual for Food Quantification [26] and the Food Weights and Portions manual [27]. Data on food and beverage intake was then converted into nutrients using the Nutrium^®^ software, which utilizes nutritional information from the food composition tables of Portugal.

UPFs were identified based on the Nova classification system [28]. For each participant, the total calories from UPF and their contribution as a percentage of the total energy intake (TEI) were calculated.

#### 2.4.5. Statistical Analysis

Data analysis was conducted using IBM SPSS Statistics for Windows^®^, version 27.0 (IBM Corp., Armonk, NY, USA). Statistical significance was set at *p* < 0.05.

The Kolmogorov–Smirnov test was used to assess the normality of data distribution. For comparisons of continuous variables between two independent groups, the independent *t*-test or the Mann–Whitney *U* test was used. For comparisons involving three or more independent groups, one-way analysis of variance (ANOVA), with Bonferroni post hoc test, or the Kruskal–Wallis test, with the Mann–Whitney test used to compare individual pairs, was performed as appropriate. The Chi-square test was used to analyze associations between categorical variables. Descriptive analysis included means and standard deviations, median and 25th (P25) and 75th percentiles (P75), as well as absolute and relative frequencies.

The Nova-UPF score was calculated by adding up the number of subgroups (ranging from 0 to 25) consumed the day prior.

For the purpose of this study, participants were divided into quintiles based on their Nova-UPF score and the contribution of calories from UPFs to their TEI.

To determine the construct validity of the Nova-UPF score, we tested the following hypotheses: (i) Nova-UPF score is higher in younger participants; (ii) the mean/median BMI is higher in the highest quintiles of the Nova-UPF score; (iii) the mean/median intake of energy, sugars, total fat, saturated fat, and sodium is higher in the highest quintiles of the Nova-UPF score; (iv) the mean/median intake of dietary fibre, potassium, monounsaturated fatty acids, and polyunsaturated fatty acids is lower in the highest quintiles of the Nova-UPF score.

Criterion validity was assessed by examining the relationship between the quintile of Nova-UPF score and the contribution of calories from UPF to total energy intake. This was performed using linear regression models with the contribution of calories from UPF to total energy intake as the dependent variable and Nova-UPF score as independent variable (continuous and divided into quintiles). The agreement between the Nova-UPF score quintile and the quintile of the percentage of energy from UPFs was evaluated using the Prevalence- and Bias-Adjusted Kappa (PABAK) coefficient [29] with quadratic weighting, which is suitable for ordinal data as it assigns greater weight to closer agreements [18]. According to established criteria, PABAK values above 0.80 indicate almost perfect agreement, values between 0.61 and 0.80 suggest substantial agreement, 0.41 to 0.60 indicate moderate agreement, 0.21 to 0.40 suggest fair agreement, and values of 0.20 or below indicate slight agreement [30]. The Pabak index was calculated using WINPEPI software (version 11.65 for Windows^®^).

## 3. Results

### 3.1. Nova-UPF Screener Adaptation

Table 1 presents the original and adapted versions of the Nova-UPF screener for the Portuguese population after the adaptation step and face-validity. The main adjustments made to the original tool included revising subgroup names for clarity, adjusting terminology, merging or dividing some subgroups, and incorporating new ones as needed to reflect common UPFs consumed by the Portuguese adult population. The final Portuguese Nova-UPF screener comprises 25 subgroups, including examples of common brands.

### 3.2. Characteristics of the Sample

Table 2 presents the sociodemographic and lifestyle characteristics of the study sample. The median age of the participants was 22 years old (P25: 20; P75:37), ranging from 18 to 63 years old. Overall, the total sample had an educational level of secondary or below (66.2%), lived in urban areas (37.3%), were non-smokers (55.7%), were non-overweight (74.8%) and were omnivores (87.6%). Compared to men, women were younger, had a higher educational level, a higher proportion of non-smokers, a lower proportion of vegetarian/plant-based dietary patterns, and had a lower BMI and physical activity (*p* < 0.05 for all).

The consumption frequency of each subgroup included in the Nova-UPF screener is detailed in Table 3. More than one third of participants reported consuming “Flavoured yoghurts and flavoured fermented milks” (47.5%), “Commercial biscuits, cookies and cerals bars” (40.8%), “Sliced bread, hamburger or hot dog buns, industrial wraps, toast, bagels or breadsticks” (36.3%), and “Chocolate bar or chocolate candies, gummy candies, sugar candies, chewing gums, toffees” (35.0%). Less than 10% of participants reported consuming “Plant-based meat substitutes” (9.6%), “Chocolate or flavoured milk in can or box” (8.9%), “Tea -based beverages” (8.9%), “Distilled spirit” (8.3%), “Frozen pizza or from fast-food restaurants” (5.4%), and “Energy and sports drinks” (2.9%).

Figure 1 presents distribution of Nova-UPF scores. The scores ranged from 0 to 12. Approximately 74% of participants scored between 1 and 5, while 22.9% scored 6 or higher.

### 3.3. Construct Validity

Table 4 describes the sociodemographic and dietary intake based on the quintiles of the Nova-UPF score. Participants in the highest quintile of the Nova-UPF score were younger (median = 21.0 years, P25–P75: 19.0–29.75) compared with those in the first (median = 25.5 years, P25–P75: 21.0–42.0) and fourth quintiles (median = 24.0 years, P25–P75: 21.0–37.0) (*p* = 0.010). A higher TEI was found for participants in the fifth quintiles compared to their counterparts (*p* < 0.001). Participants in the fifth quintile of the Nova-UPF score had a lower median intake of monounsaturated fatty acids (median = 5.9% TEI, P25–P75: 3.44–9.46) compared with those in the first (median = 8.7% TEI, P25–P75: 5.79–11.95) and second (median = 8.1% TEI, P25–P75: 5.85–11.00) quintiles (*p* = 0.001), and lower intakes of polyunsaturated fatty acids (median = 2.8% TEI, P25–P75: 1.63–3.74) compared with the first (median = 4.0% TEI, P25–P75: 2.62–5.68), second (median = 4.1% TEI, P25–P75: 2.85–5.65), and third quintiles (median = 4.0% TEI, P25–P75: 2.22–5.13) (*p* < 0.001). Potassium intake was also significantly lower among participants in the highest quintile (median = 1694.9 mg/day (1128.3; 2407.9) relative to those in the first (median = 2555.4 mg/day P25–P75: 1865.3–3410.3), second (median = 2289.0 mg/day P25–P75: 1603.0–2921.8), and fourth quintiles (median = 2394.1 mg/day, P25–P75: 1738.1–3201.7). A higher dietary fibre intake was found for the participants in the first quintile (median = 20.6 g/day, P25–P75: 13.43–27.78) compared to the third (median = 16.6 g/day, P25–P75: 9.90–21.3) and fifth quintiles (median = 17.1 g/day, P25–P75: 11.30–22.78) (*p* = 0.024).

No differences were found for BMI, carbohydrates, total fat, saturated fat, and sodium intake.

### 3.4. Criterion Validity and Agreement

The average contribution of UPF to TEI was 27.2% (CI 95%: 24.90–29.40) for the total sample. Participants in the first quintile had a mean energy contribuition of UPF of 10.5% (CI 95%: 8.54–12.46) and those in the fifth quitile had a mean energy contribuition around half of the TEI (Table 5). The regression linear models showed a positive association with UPF consumption and Nova-UPF score (Nova-UPF score as continuous varaiable: B = 6.224, *p*-value < 0.001; Quintiles of Nova-UPF scores: B = 9.269 *p*-value for linear trend < 0.001).

Table 6 displays the distribution of participants based on their classification of quintiles of dietary share of UPF (according to the 24 h dietary recall) and quintiles of Nova-UPF score. The results show a nearly perfect agreement between the two criteria (PABAK coefficient = 0.86).

## 4. Discussion

The present study aimed to adapt and validate the Nova-UPF screener for Portuguese adults to assess consumption of UPFs. The findings showed a significant positive association between the Nova-UPF score and the percentage of energy intake from UPFs, as estimated by a 24 h dietary recall. Additionally, there is a near-perfect agreement between the quintiles of Nova-UPF score and the quintile of UPF energy contribution.

In this study, UPFs accounted for 27.2% of the TEI, which aligns with national data from the IAN-AF 2015–2016 (24%) [9], and is comparable to estimates reported for other European countries [31,32]. Compared with previous Nova-UPF screener validation studies, the contribution of UPFs in our sample was higher than that observed in Senegal (17.4%) [17], and similar to that reported in Colombia (27.3%) [20]. Consumption of flavoured yoghurts, commercial biscuits and cookies, industrial bread, and chocolates/candies was reported by more than one-third of the participants. These findings show a slightly different consumption pattern compared to the validation studies of the Nova-UPF screener conducted in Brazil, Senegal, Ecuador, India, and Colombia, highlighting cross-country differences likely reflecting the cultural and contextual diversity of population dietary patterns. For instance, in the Brazilian sample, over 30% of participants reported consuming margarine, “loaf, hot dog or hamburger bread”, and “regular or diet soft drinks” [15]. In Senegal, the most frequently reported UPFs included “bouillons, dipping sauces, vinaigrettes, and industrial salad dressings” and “industrial mayonnaise, ketchup or mustard” [17]. In Ecuador, common UPFs were “sliced or industrial bread”, “tea or coffee prepared from powdered mixes”, “mayonnaise, ketchup or mustard” and “chocolate, and candies” [18]. In India, the most consumed UPFs included “packaged and branded biscuits, cream biscuits, cookies, cream puffs/rolls”, “packaged and branded ketchup, chutneys/instant chutney powders/tastemakers, packaged and branded pickles, sauces (like pasta-pizza sauce), instant gravies/curries/pastes”, and “packaged and branded bread” [19].

In this study, the Nova-UPF score ranged from 0 to 12, with scores of 1 (14%), 2 (15.6%), and 3 (19.4%) being the most common values. Similar score ranges were reported in validation studies conducted in India [19] and Ecuador [18]. In contrast, the Brazilian [15] and Senegalese [17] validations showed lower score ranges, from 0 to 9 and 0 to 8, respectively. These differences may partly reflect cultural and dietary variations across populations but could also be influenced by the age distribution of the samples. In our study, the average age of participants was lower compared to other validation studies. This may have contributed to the higher Nova-UPF scores observed, as younger adults tend to consume more UPFs [12,33]. Additionally, we found that participants in the highest quintile of the Nova-UPF score were younger compared to those in the first quintile. Data from the IAN-AF 2015–2016 found that younger ages were significantly associated with higher UPF consumption in both sexes [12]. Consistent with this, a systematic review that included 55 nationally representative studies across 32 countries found, among other sociodemographic variables, that age was inversely and independently associated with UPF intake [33]. This is concerning, as dietary habits established at younger ages tend to persist into adulthood, potentially increasing the long-term risk of diet-related chronic diseases [34].

A stronger concordance was observed between the quintiles of Nova-UPF score and the quintiles of the percentage of UPF-derived energy among individuals in the extreme quintiles (first and fifth), suggesting that the tool more accurately distinguishes between low and high UPF consumers. Similar findings were seen in the Brazilian [15] and Indian [19] populations. Moreover, a near-perfect agreement was observed between the quintiles of Nova-UPF score and the quintiles of UPF contribution to TEI, which was comparable to the validation studies conducted in Senegal (PABAK = 0.84) [17], India (PABAK = 0.85) [19], and Ecuador (PABAK = 0.82) [18], and higher than that observed for the original tool (PABAK = 0.67) [15]. These findings highlight that the Nova-UPF screener can accurately classify individuals according to their UPF consumption, reinforcing its applicability for dietary assessment and public health research.

Overall, the construct validity of the Nova-UPF screener was largely supported by our findings. In line with our predefined hypotheses, participants in the highest quintiles of the Nova-UPF score were younger and had greater TEI but lower consumption of monounsaturated and polyunsaturated fatty acids, potassium, and dietary fibre compared with those in the lowest quintile. These results align with evidence from a meta-analysis of nationally representative surveys examining UPF consumption and diet quality, which demonstrated that higher UPF intake is associated with poorer nutritional quality [35]. In that meta-analysis study, UPF consumption was positively correlated with free sugars, total fats, and saturated fats, and inversely correlated with fibre, protein, potassium, zinc, magnesium, and vitamins A, C, D, E, B12, and niacin. Likewise, data from the I. Family study [10], conducted across eight European countries, showed that among adults, a higher dietary share of UPFs was positively associated with TEI and negatively associated with dietary fibre intake. Consistent with our results, that study also reported no significant associations with total and saturated fat intake. Taken together, Nova-UPF seems to reflect established associations between UPF intake and key nutritional indicators reinforcing its potential utility. On the other hand, no significant association was found between Nova-UPF score quintiles and BMI. Similarly, studies conducted among university students [36] and adult populations [37,38] have shown that the proportion of energy derived from UPFs was not higher among obese individuals compared to their non-obese counterparts. Several factors may account for this finding. First, the high proportion of participants classified as non-overweight may have limited the ability to detect significant differences in BMI across Nova-UPF score quintiles. While nearly a quarter (24.3%) of non-overweight participants fell into the highest Nova-UPF quintiles, a higher proportion of overweight participants (39.2%) were found in the lowest quintiles. Additionally, body weight and height were self-reported, which are prone to measurement error and may led to BMI misclassification, as self-reported anthropometric data often underestimate body weight and, consequently, BMI [39,40]. Moreover, evidence suggests that the relationship between UPF consumption and adiposity may not be fully captured by BMI alone, particularly in younger or predominantly non-overweight populations, as BMI does not adequately reflect body fat distribution or early metabolic changes [41]. 

The cross-country adaptation of the Nova-UPF screener highlights both its flexibility and the challenges inherent in its use in different food environments. Although the tool is based on the NOVA classification, ensuring content validity requires adapting food examples and UPF subgroups to align with locally available and commonly consumed products, all while maintaining conceptual equivalence with the original instrument. Previous validation studies have shown that this balance is essential to preserve respondent understanding and measurement accuracy without compromising cross-country comparability [17,18,19,20]. Our adaptation process, which is based on national dietary data and expert consensus, provides a pragmatic framework that can support the application of the Nova-UPF screener in other countries and facilitate international monitoring of UPF consumption trends.

This study highlights some limitations and strengths that should be considered when interpreting the findings. Firstly, the validation of the Nova-UPF screener was conducted using a 24 h dietary recall, which is inherently prone to recall and reporting biases. However, the use of the five-stage multiple-pass method likely enhanced the completeness and accuracy of dietary reporting [25]. Additionally, relying solely on a single 24 h dietary recall may lead to misclassifying typical consumption of UPF, as the 24 h dietary recall only captures short-term intake and may not accurately reflect habitual dietary patterns. However, our objective was not to estimate usual UPF intake, but rather to evaluate the screener’s ability to rank individuals according to UPF exposure. The 24 h recall is widely used in validation studies as a reference method for dietary screeners, particularly when the goal is to rank individuals based on intake rather than to estimate usual consumption [42]. Furthermore, since the Nova-UPF screener also assesses consumption on the previous day, employing multiple 24 h recalls could potentially weaken the level of agreement between the two instruments by introducing dietary data from days not covered by the screener. Moreover, the observed associations between the Nova-UPF score and UPF energy contribution suggest that the screener performs adequately. Secondly, the validation conducted using a convenience sample consisted of mostly of young participants with a secondary education level or lower, residing in urban areas, non-smokers, and individuals who were not overweight. Although this limits the generalizability of the results, the UPF consumption levels observed in this study were comparable to those reported in the nationally representative IAN-AF survey. Despite these limitations, the Nova-UPF screener demonstrated notable strengths, including being simple, time-efficient, and easy to administer while accurately capturing UPF consumption among adults. Moreover, its practicality makes it a valuable tool for large-scale epidemiological studies and for monitoring trends in UPF intake over time, contributing to the evaluation and development of nutrition-related public health policies.

## 5. Conclusions

This study provides evidence that the Nova-UPF screener is a valid instrument for assessing the consumption of UPFs among Portuguese adults. The screener’s score accurately reflects the contribution of UPFs to total daily energy intake, supporting its accuracy and applicability in nutritional epidemiology. Due to its simplicity, rapid administration, and ease of interpretation, the Nova-UPF screener represents a valuable tool for both research and practice, allowing for the efficient assessment of dietary patterns characterized by UPF intake. Moreover, its use can contribute to monitoring trends in UPF consumption, helping to identify at-risk populations and inform the development of public health policies and interventions aimed at improving diet quality and reducing the burden of diet-related non-communicable diseases.

## Figures and Tables

**Figure 1 nutrients-18-00090-f001:**
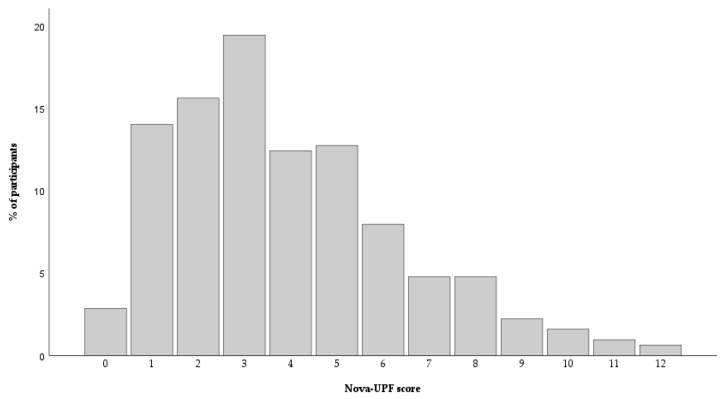
Distribution of the Nova-UPF score for the total sample (n = 314).

**Table 1 nutrients-18-00090-t001:** Original and adapted version of Nova-UPF screener.

Original (Brazil)	Adapted
Regular or diet soda	Soft drinks, fruit drink prepared from a powdered mix, aromatized waters, tonic waters, packaged lemonade
Canned or bottled fruit juice	Fruit/vegetable-based juices in a can or box
Powered drink mix	[COMBINED] “Fruit drink prepared from a powdered mix” was removed and added to subgroup “Soft drinks, fruit drink prepared from a powdered mix, aromatized waters, tonic waters, packaged lemonade”
Chocolate drink	Chocolate or flavoured milk (with and without lactose)
Tea-based beverages	Tea-based beverages
Fruit- or chocolate-flavoured yoghurt	Flavoured yoghurts, solid or liquid (with or without lactose) or flavoured fermented milks (with or without lactose)
-	[NEW] Plant-based drinks (soy, rice, almond, among others), plant-based alternatives to yoghurt and fermented milk
-	[NEW] Energy and sports drinks
-	[NEW] Distilled spirit (includes all alcoholic beverages other than cider, beer or wine, e.g., spirits, vodka, whisky)
Sausage, hamburger, or nuggets	Sausages, hamburger meat, nuggets, fish fingers, surimi, industrial pâtés
Ham, salami, or mortadella	Ham, salami, mortadella, paio sausage, chorizo
Loaf, hot dog, or hamburger bread	Sliced bread, hamburger or hot dog buns, industrial wraps, toast, bagels or breadsticks
Margarine	Margarine, chocolate/cocoa and hazelnut spreads
Mayonnaise, ketchup, or mustard	[COMBINED] Ready-made sauces (any type, including salad dressings), mustard, ketchup and mayonnaise, stock cubes
Ready-made salad sauce	
Frozen fries, either frozen or from fast-food restaurants	Frozen French fries or from fast-food restaurants and frozen and instant mashed potatoes
Pizza, either frozen or from fast-food restaurants	Frozen pizza or from fast-food restaurants
Instant noodles or packaged soup	Ready-to-eat foods (burgers, fast food, packaged salads with dressings or from fast-food chain)
Frozen lasagna or other frozen ready-made meal	Pre-cooked foods (frozen and canned/packaged meals, noodles)
-	[NEW] Spreadable/melted cheese, flavoured *petit suisse*
-	[NEW] Plant-based meat substitutes (vegetable burgers, vegetable sausages, seitan, industrial vegetarian pâté)
Packaged snacks, shoestring potatoes or crackers	Crackers, potato chips, packaged savoury/sweet popcorn, and packaged fried snacks
Biscuits with or without filling	[COMBINED] Commercial biscuits and cookies with or without filling, cereal bars
Cereal bar	
Packaged cake	Packaged cakes, pies, croissants and other pastries, with or without cream
Ice cream or popsicle	Ice cream or ice lolly (not homemade or artisanal)
Chocolate bar or bonbon	Chocolate bar or chocolate candies, gummy candies, sugar candies, chewing gums, toffees
Breakfast cereals	Sugary breakfast cereals

**Table 2 nutrients-18-00090-t002:** Sociodemographic and lifestyle characteristics of the total sample and by sex.

	Total (n = 314)	Women (n = 197)	Men (n = 117)	*p*-Value
Age, years	22.0 (20.0; 37.0)	22.0 (20.0; 28.0)	30.0 (21.0; 45.0)	<0.001
Marital Status, n (% married)	70 (22.3)	30 (15.2)	40 (34.2)	<0.001
Education level, n (%)				
Basic	27 (8.6)	10 (5.1)	17 (14.5)	0.009
Secondary	181(57.6)	114 (57.9)	67 (57.3)	
College/university	106 (33.8)	73 (37.1)	33 (28.2)	
Urbanization degree, n (%)				
City	117 (37.3)	79 (40.1)	38 (32.5)	0.373
Peripheral city	103 (32.8)	63 (32.0)	40 (34.2)	
Village/community settlement	94 (29.9)	55 (27.9)	39 (33.3)	
Smoking habits, n (%)				
Non-smokers	175 (55.7)	120 (60.9)	55 (47.0)	0.026
Former smokers	97 (30.9)	57 (28.9)	40 (34.2)	
current smokers	42 (13.4)	20 (10.2)	22 (18.8)	
BMI. kg/m^2^	23.7 (21.1; 25.1)	22.4 (20.7; 24.7)	24.1 (22.5; 25.9)	<0.001
Weight Status, n (%)				
Non-overweight	235 (74.8)	155 (78.7)	80 (68.4)	0.042
Obese	79 (25.2)	42 (21.3)	37 (31.6)	
Dietary pattern, n (%)				
Omnivore	275 (87.6)	166 (84.3)	109 (93.2)	0.018
High animal-based diet (paleo + ketogenic)	6 (1.9)	3 (1.5)	3 (2.6)	
Vegetarian/Plant-based	33 (10.5)	28 (14.2)	5 (4.3)	
Physical Activity (min/week)	370.0 (170.0; 760.0)	325.0 (175.0; 595.0)	465.0 (167.5; 960.0)	0.012

Data are median (25th percentile; 75th percentile) for continuous variables; BMI, body mass index.

**Table 3 nutrients-18-00090-t003:** Consumption frequency (%) of the subgroups included in the Nova-UPF screener for the total sample (n = 314).

Ultra-Processed Foods Subgroups	%
Flavoured yoghurts, solid or liquid (with or without lactose) or flavoured fermented milks (with or without lactose)	47.5
Commercial biscuits and cookies with or without filling, cereal bars	40.8
Sliced bread, hamburger or hot dog buns, industrial wraps, toast, bagels or breadsticks	36.3
Chocolate bar or chocolate candies, gummy can-dies, sugar candies, chewing gums, toffees	35.0
Ham, salami, mortadella, paio sausage, chorizo	30.9
Spreadable/melted cheese, flavoured *petit suisse*	28.7
Margarine, chocolate/cocoa and hazelnut spreads	27.4
Packaged cakes, pies, croissants and other pastries, with or without cream	26.8
Ready-made sauces (any type, including salad dressings), mustard, ketchup and mayonnaise, stock cubes	22.0
Soft drinks, fruit drink prepared from a powdered mix, aromatized waters, tonic waters, packaged limonade	19.4
Frozen French fries or from fast-food restaurants and frozen and instant mashed potatoes	19.1
Sausages, hamburger meat, nuggets, fish fingers, surimi, industrial pâtés	17.5
Plant-based drinks (soy, rice, almond, among others), plant-based alternatives to yoghurt and fermented milk	15.9
Fruit/vegetable-based juices in a can or box	15.0
Branded ice cream or ice lolly (not homemade or artisanal)	12.4
Crackers, potato chips, packaged savoury/sweet popcorn, and packaged fried snacks	11.8
Ready-to-eat foods (burgers, fast food, packaged salads with dressings or from fast-food chain)	11.1
Sugary breakfast cereals	10.8
Pre-cooked foods (frozen and canned/packaged meals, noodles)	10.2
Plant-based meat substitutes (vegetable burgers, vegetable sausages, seitan, industrial vegetarian pâté)	9.6
Chocolate or flavoured milk (with or without lactose)	8.9
Tea-based beverages	8.9
Distilled spirit (includes all alcoholic beverages other than cider, beer, or wine, e.g., spirits, vodka, whisky)	8.3
Frozen pizza or from fast-food restaurants	5.4
Energy and sports drinks	2.9

**Table 4 nutrients-18-00090-t004:** Age, body mass index, and dietary intake according to quintiles of the Nova-UPF score.

	Nova-UPF Score Quintiles					*p*-Value
	Q1 (n = 102)	Q2 (n = 61)	Q3 (n = 39)	Q4 (n = 40)	Q5 (n = 72)	
Age ^a^, years	25.5 (21.0; 42.0) ^d,g^	22.0 (20.0; 32.0) ^c^	22.0 (20.0; 30.0)	24.0 (21.0; 37.0) ^g^	21.0 (19.0; 29.75) ^c,f^	0.010
BMI ^a^, kg/m^2^	23.8 (21.45; 25,89)	21.5 (20.49; 24.80)	23.5 (21.34; 24.98)	23.6 (21.23; 25.46)	22.4 (20.71; 24.75)	0.063
Dietary intake						
TEI ^a^, kcal/day	1715.9 (1304.5; 2147.1) ^g^	1789.6 (1453.4; 2276.3) ^e,g^	1593.5 (1286.4; 1790.0) ^d,f,g^	1823.6 (1421.3; 2331.1) ^e,g^	1912.3 (1624.0; 2557.1) ^c,d,e,f^	<0.001
Carbohydrates ^b^, %TEI	42.8 ± 10.46	44.3 ± 8.62	46.3 ± 8.57	46.2 ± 8.43	45.3 ± 8.93	0.155
Sugars ^b^, %TEI	13.7 ± 6.53	13.1 ± 5.16 ^e^	16.7 ± 6.03 ^d^	15.4 ± 5.88	13.8 ± 6.49	0.030
Total fat ^b^, %TEI	30.6 ± 8.95	30.8 ± 7.75	30.7 ± 8.44	29.2 ± 6.52	32.8 ± 7.86	0.209
Saturated fat ^a^, %TEI	8.6 (6.29; 9.97)	8.9 (6.85; 10.93)	7.5 (5.43; 10.12)	8.2 (6.64; 9.65)	9.7 (6.88; 12.22)	0.070
MUFA ^a^, %TEI	8.7 (5.79; 11.95) ^g^	8.1 (5.85; 11.00) ^g^	7.1 (4.70; 10.11)	8.1 (4.68; 10.12)	5.9 (3.44; 9.46) ^c,d^	0.001
PUFA ^a^, %TEI	4.0 (2.62; 5.68) ^g^	4.1 (2.85; 5.65) ^f,g^	4.0 (2.22; 5.13) ^g^	3.2 (2.10; 4.20) ^d^	2.8 (1.63; 3.74) ^c,d,e^	<0.001
Sodium ^a^, mg/day	2781.3 (2006.2; 3685.0)	2924.6 (2186.2; 3747.7)	2285.5 (1611.0; 2854.4)	2610.9 (1892.4; 3343.8)	2860.8 (1901.7; 3772.9)	0.081
Potassium ^a^, mg/day	2555.4 (1865.3; 3410.3) ^e,g^	2289.0 (1603.0; 2921.8) ^e,g^	1895.1 (1334.5; 2494.5) ^c,d,f^	2394.1 (1738.1; 3201.7) ^e,g^	1694.9 (1128.3; 2407.9) ^c,d,f^	<0.001
Dietary fibre ^a^, g/day	20.6 (13.43; 27.78) ^e,g^	19.1 (14.15; 24.25)	16.6 (9.90; 21.3) ^c^	16.0 (12.58; 20.75)	17.1 (11.30; 22.78) ^c^	0.024

Data are median (25th percentile; 75th percentile) or mean ± standard deviation. ^a^ Analysis by Kruskal–Wallis; ^b^ Analysis by one-way analysis of variance; ^c^ *p* < 0.05, compared with the quintile 1; ^d^ *p* < 0.05, compared with the quintile 2; ^e^ *p* < 0.05, compared with the quintile 3; ^f^ *p* < 0.05, compared with the quintile 4; ^g^ *p* < 0.05, compared with the quintile 5. BMI, body mass index; MUFA, monounsaturated fatty acids; Q: quintile; PUFA, polyunsaturated fatty acids; TEI, total energy intake; UPF, ultra-processed foods; Q1: 0–2; Q2: 3; Q3: 4; Q4: 5; Q5: ≥6.

**Table 5 nutrients-18-00090-t005:** Dietary share of ultra-processed foods based on 24 h dietary recall according to the Nova-UPF score quintiles (n = 314).

Nova-UPF Score	n	Dietary Share of Ultra-Processed Foods (% of TEI) Mean (95% CI)
Q1 [0–2]	102	10.5 (8.54–12.46)
Q2 [3]	61	22.1 (19.62–24.49)
Q3 [4]	39	32.8 (28.20–37.33)
Q4 [5]	40	29.5 (25.83–33.19)
Q5 [≥6]	72	50.7 (45.98–55.43)

PABAK (prevalence-adjusted bias-adjusted Kappa) = 0.86. CI, confidence interval; Q, quintile; TEI, total energy intake.

**Table 6 nutrients-18-00090-t006:** Quintiles of dietary share of ultra-processed foods based on 24 h dietary recall according to quintile of Nova-UPF score (n = 314).

	Nova-UPF Score Quintiles					
Quintiles of Dietary Share of UPF (% of TEI)	Q1 [0–2]	Q2 [3]	Q3 [4]	Q4 [5]	Q5 [≥6]	Total
Q1 (≤9.74)	18.8	1.3	0	0	0	20.1
Q2 (9.75–19.22)	9.6	7.0	1.6	1.9	0	20.1
Q3 (19.23–26.78)	2.9	6.1	4.1	5.1	1.9	20.1
Q4 (26.78–42.76)	1.0	4.1	3.2	4.1	7.6	20.1
Q5 (≥42.77)	0.3	1.0	3.5	1.6	13.4	19.7
Total	32.5	19.4	12.4	12.7	22.9	100

PABAK coefficient = 0.86. Q, quintile; TEI, total energy intake; UPF, ultra-processed foods.

## Data Availability

The raw data supporting the conclusions of this article will be made available by the authors on request.

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
