# Peer review of "Nutrients2026, 18(1), 90;https://doi.org/10.3390/nu18010090"

_nutrients, 2025, doi:10.3390/nu18010090_

Round 1

Reviewer 1 Report

Comments and Suggestions for Authors

This study evaluated the validity of the adapted Nova-UPF screener in the Portuguese population and demonstrated good validity and agreement with the Nova classification. The study provides valuable insights into the applicability of a Nova-specific food questionnaire and its potential use for improving the classification and capture of ultra-processed food consumption. Below are some comments for the authors’ consideration:

Major comments:

Line 115-119: For those who are not familiar with the screener tool, the author should explain this in greater details, including the intended purpose of the tool, how the screener assigns scores to participants, whether the score represents a quantitative measure or a ranking metric etc.

Table 4: The use of superscript is really confusing, the author should decide on use either letters or symbols across the board.

The authors outline four hypotheses to assess construct validity; however, the Discussion does not explicitly state which hypotheses were supported and which were not. This should be clearly acknowledged and discussed. In particular, the lack of association between the Nova-UPF score and BMI warrants further discussion.

Lines 341–356: This paragraph does not add anything substantial to the discussion or conclusions. I recommend either removing this paragraph completely or significantly shortening it, as it currently reads repetitive and does not meaningfully contribute to the manuscript’s key messages.

The authors should also discuss the potential implications of cross-country adaptation of this screener, including challenges and key considerations when adopting it in other countries. Given that the manuscript devotes a substantial portion of the text to describing the process of adapting the screener to the local population, these points should also be explicitly acknowledged and discussed.

Furthermore, the author should acknowledge that the use of a single 24-hour dietary recall reflects short-term intake and may not capture habitual dietary patterns, which could affect the assessment of usual UPF consumption.

Minor, grammar/typos:

Line 82: "it is essential finds alternative" should be "it is essential to find.."

Line 95: "it is a burning issue to adapt" sounds awkward, maybe rephrase it to "an urgent issue" or "an important issue"

Line 157: "to standardize procedures and ensure consistency data collection.", it should be "ensure consistency in data collection".

Line 165: "Sociodemographic factors as age, educational level, sex, marital status" - Sociodemographic factors such as age? or use "including" age.

Line 308: should be "a positive association" not "an".

Line 309: continous should be continuous.

Reviewer 2 Report

Comments and Suggestions for Authors

The manuscript presents the validation of the Nova-UPF Screener for the Assessment of Ultra-Processed Food Intake in Portuguese Adults. The subject is valid and current. The methodology is well selected, and the tool is of great importance due to the challenges posed by ultra-processed foods (UPFs). The discussion presents the Portuguese version of the tool in comparison with other contemporary available tools.

I have only minor comments on the Methods and Results.

Since the Delphi methodology was used in the first step of the process, the reader would benefit from presenting this stage in more detail (e.g., how was the agreement of experts defined?; have the experts reached consensus in all statements?; how many rounds were there?). In general, the Delphi methodology is poorly visible in the manuscript.

Face validity: an example of “minor wording and formatting adjustments” would be welcome to illustrate this procedure.

Statistical analysis: in rows 213-214, “Descriptive analysis included means and standard deviations, median and 25th (P25) and 75th percentiles (P75),” whereas it does not fully correspond to further text. For example, in row 251, only the mean age and percentiles are presented. The same applies to Table 2.

There appears to be some discrepancy between the Results (row 272: Approximately 60% of participants scored between 2 and 5) and the Discussion (row 357: the Nova-UPF score ranged from 0 to 12, with scores of 1, 2, and 3 being the most common values).
